# Increased serum resistin but not G-CSF levels are associated in the pathophysiology of major depressive disorder: Findings from a case-control study

Smaranika Rahman[1◉], Amena Alam Shanta[1◉], Sohel Daria[1], Zabun Nahar[1], Mohammad Shahriar[1], MMA Shalahuddin Qusar[2], Sardar Mohammad Ashraful Islam[1], Mohiuddin Ahmed Bhuiyan[1], Md. Rabiul Islam[1]*

1 Department of Pharmacy, University of Asia Pacific, Farmgate, Dhaka, Bangladesh, 2 Department of Psychiatry, Bangabandhu Sheikh Mujib Medical University, Shahabagh, Dhaka, Bangladesh

◉ These authors contributed equally to this work.
* robi.ayaan@gmail.com

**Data Availability Statement:** All relevant data are within the manuscript and its Supporting information files.

## Abstract

### Background

Many studies have predicted major depressive disorder (MDD) as the leading cause of global health by 2030 due to its high prevalence, disability, and illness. However, the actual pathophysiological mechanism behind depression is unknown. Scientists consider alterations in cytokines might be tools for understanding the pathogenesis and treatment of MDD. Several past studies on several inflammatory cytokine expressions in MDD reveal that an inflammatory process is activated, although the precise causes of that changes in cytokine levels are unclear. Therefore, we aimed to investigate resistin and G-CSF in MDD patients and controls to explore their role in the pathogenesis and development of depression.

### Methods

We included 220 participants in this study. Among them, 108 MDD patients and 112 age-sex matched healthy control (HCs). We used DSM-5 to evaluate study participants. Also, we applied the Ham-D rating scale to assess the severity of patients. Serum resistin and G-CSF levels were measured using ELISA kits (BosterBio, USA).

### Results

The present study observed increased serum resistin levels in MDD patients compared to HCs (13.82 ± 1.24ng/mL and 6.35 ± 0.51ng/mL, p <0.001). However, we did not find such changes for serum G-CSF levels between the groups. Ham-D scores showed a significant correlation with serum resistin levels but not G-CSF levels in the patient group. Furthermore, ROC analysis showed a fairly predictive performance of serum resistin levels in major depression (AUC = 0.746).

**Funding:** The author(s) received no specific funding for this work.

**Competing interests:** The authors have declared that no competing interests exist.

## Conclusion

The present study findings suggest higher serum resistin levels are associated with the pathophysiology of MDD. This elevated serum resistin level may serve as an early risk assessment indicator for MDD. However, the role of serum G-CSF in the development of MDD is still unclear despite its neuroprotective and anti-inflammatory effects.

## Introduction

Major depressive disorder (MDD) is a complex and devastating mental health disorder across the world. Depressive people experience anhedonia in which they do not find peace in activities that were pleasurable once; usually characterized by the persistent occurrence of depressed mood, low self-esteem, worse sleep or appetite, and even thought of suicide or death [1]. Clinicians define major depression as when the depressive symptoms accompanied by other physical changes last for at least two weeks. These changes include psychomotor retardation or agitation, alteration in appetite, body weight, sleep pattern, and sustained fatigue depending on the severity and number of episodes [2]. The WHO has classified MDD as the third largest cause of disease burden since 2008. The authority expects depression to be the leading cause of global disease by 2030 [3]. Worldwide, about 280 million people suffer from depression, and women are more susceptible to developing depressive symptoms than men [4]. Approximately 20% of the adult population worldwide may experience depressive episodes at some point in their lives, and about 80% of MDD patients may have at least one more episode [5,6]. Depression is often incorrectly diagnosed or misdiagnosed due to a lack of recourses, skilled healthcare practitioners, and social stigma associated with mental disorders [4]. However, the most significant impediments to the successful management of depression are the lack of appropriate quantitative early risk assessment markers and inaccurate measurements. MDD is currently diagnosed mostly through clinical examination, patient self-report, and subjective evaluation of depressed symptoms [7].

MDD is a multifaceted, multi-etiological, unpredictable disease condition with a variety of symptoms. There is no single reason behind the onset of depression. A complex interaction among genetic, biochemical, nutritional, environmental, and psychological variables causes depression [8–13]. However, the pathophysiology of MDD is still inconclusive. Several hypotheses are potentially associated with the pathophysiology of MDD. Among them, cytokine alterations [14], inflammatory responses [15], the hypothalamic-pituitary-adrenal (HPA) axis activation [12], serotonin mechanism [16], catecholamine synthesis [17], and neurotrophic actions [18] are noticeable. Cytokines are proteins in nature, and immune cells secrete them. Therefore, they can cause local and systemic immune responses and induce inflammation [14]. The activated microglia produce pro-inflammatory cytokines during inflammation. These increased cytokine levels hyperactivate the HPA axis. Therefore, cortisol, adrenocorticotropic hormone, and corticotrophin-releasing hormone levels increase in the human body [19]. They also boost the activity of the 2-indoleamine, 3-dioxygenase enzymes. Altogether they cause neurodegeneration and ultimately MDD development [20].

Resistin is a pro-inflammatory adipocytokine [21]. Inflammatory cells such as macrophages can produce it [22]. It can cause the expression of many cytokines in the human body [23–25]. Cytokine expression inhibits the synthesis of dopamine and noradrenaline in the central nervous system by the hypothalamus. Therefore, reduced intra-synaptic monoamine levels may develop depressive symptoms [26–28]. However, G-CSF involves in pro-and anti-inflammatory actions [29–31]. The neuroprotective properties of G-CSF receptors can show anti-

inflammatory effects on dopaminergic neurons during the neurodegenerative process in the brain [32–35]. Many studies have revealed increased serum resistin and G-CSF levels in MDD patients [36,37]. But Papacostas et al. reported that serum resistin levels decreased in patients suffering from depression. Also, they did not notice any significant change in serum resistin levels between MDD and healthy controls (HCs) in their replication studies [28]. However, some other studies have failed to show significant changes in the serum levels of resistin and G-CSF in MDD patients compared to HC [38,39]. Therefore, the therapeutic significance of the relationship between serum resistin and G-CSF levels with depression is still unclear. Thus, we aimed to evaluate the serum resistin and G-CSF of MDD patients in a case-control study to find their relationship with the pathogenesis of depression.

## Materials and methods

### Study population

We thought the exposed percentage and alpha risk as 10% and 5%, respectively. Also, we considered 90% statistical power and an odds ratio of 2 for this 1:1 matched case-control study. According to the above estimates, the hypothetical sample size was 252 (126 MDD patients and 126 HCs). Therefore, we included 108 MDD patients and 116 HCs from the native Bangladeshi population. We recruited MDD patients from a tertiary care teaching hospital in the capital city of Bangladesh. Before participation, we obtained informed consent from each participant of the study. A psychiatrist assessed MDD patients based on the DSM-5 criteria for this study. We recruited age-sex matched HCs from different parts of the capital city in Bangladesh. We interviewed all the study participants using the same structured questionnaire to rule out any previous or current psychotic condition defined by the DSM-5. Also, we applied the Ham-D rating scale to assess the severity of MDD patients suffering from depression. We included depressive patients who had a Ham-D score of seven or higher. This study excluded subjects from the present study who had other comorbid mental health disorders. Also, we considered cardiovascular diseases, renal or liver disorders, excessive obesity, abnormal body mass index (BMI), autoimmune diseases, infectious diseases, uncontrolled endocrine diseases as additional exclusion criteria for this study participants. This study also excluded participants who were addicted to alcohol or other substances abuse. The socio-demographic characteristics of both patients and healthy individuals such as height, weight, and BMI were examined and recorded appropriately using a predesigned questionnaire. We performed all investigations following the principle described in the declaration of Helsinki, Seoul, Korea, version 2008.

### Blood sample collection

We collected a 5ml blood sample from the cephalic vein of each participant. We placed the collected blood sample in a falcon tube and allowed it for coagulation for half an hour at room temperature. Then, the coagulated blood samples were centrifuged at 3000rpm for 15 minutes. We collected transparent serum samples from the upper part of falcon tubes. Finally, the serum samples were aliquoted into the polypropylene tube and preserved at -80˚C until further investigation.

### Measurement of serum cytokine levels

We measured serum resistin and G-CSF levels using ELISA kits following the instructions supplied by the kit manufacturer (BosterBio, USA). At first, we placed 100μl of sample and standard solutions into the appropriate wells in a 96-well microtiter plate. We incubated the plate for two hours at room temperature. Then we removed the liquid from the wells. A 100μl of the

biotinylated anti-human antibody of each cytokine was added to the corresponding wells and mixed thoroughly. We sealed the plates again and incubated them for another 60 minutes at 37˚C temperature. After this incubation, we aspirated the contents and wells rinsed three times with 300μl of wash buffer. Then we added 100μl of the avidin-biotin-peroxidase complex into the wells and incubated at 37˚C temperature for another 30 minutes. After aspirating the contents of each well once more, we rinsed them five times with 300μl of wash buffer. We incubated the plates again at room temperature for half an hour in the dark place after adding 90μl of the color development reagent. After this incubation, we added 100μl stop solution into each well to stop the reaction process. We measured the absorbance at 450 nm within 30 minutes after stopping the reaction. We calculated serum resistin and G-CSF levels as pg/mL. The minimum detection value for serum resistin and G-CSF were <1pg/ml and <0.5pg/ml, respectively. The researcher team who conducted the assays was blind to the identities of the sample or any clinical data. Additionally, we performed all assays by the same research team to avoid interpersonal heterogeneity in the study results of measured cytokines.

## Statistical analysis

We used Microsoft excel 2016 and SPSS software (version 25) for data processing and analysis. Independent sample t-tests and Pearson's correlation coefficient test were used to compare the study parameters between MDD patients and HCs. Also, we applied Fisher's exact test for categorical variables. Pearson correlation coefficient test was used to find the association between laboratory findings and Ham-D scores among MDD patients. We used boxplot graphs to show the changes in serum cytokine levels in MDD patients and HCs. In the patient group, we used scatter plot graphs to illustrate associations of serum levels of resistin and G-CSF with Ham-D scores. Also, we performed ROC curve analysis to predict the diagnostic performance of altered cytokine in major depression. We presented the data as mean values±standard error of the mean (SEM). We considered significant results from the statistical analyses if the p-values indicated as 0.05 or lower.

## Ethics

The ethical review committee of Bangabandhu Sheikh Mujib Medical University approved the research protocol (No. BSMMU/2019/3507). We briefed about the objective of this study to all participants and obtained written consent from them. We performed all investigations following the Declaration of Helsinki.

## Result

### Sociodemographic profiles of the study population

We categorized all the participants in this study according to their biophysical features and sociodemographic profiles (Table 1).

Both cases and controls were similar in terms of their age (MDD patients: 32.15±0.88 years, HCs: 33.67±0.89 years; p = 0.226), BMI (MDD patients: 25.03±0.46 kg/m$^2$, HCs: 24.85±0.38 kg/m$^2$; p = 0.761), and smoking history (MDD patients/HCs: 34.25%/33.04%; p = 0.848). Participants with female sex were higher in MDD patients and HCs than males. The majority portion of the study population were nonsmokers and belonging to the medium economic class. We noticed a significant portion of MDD patients were young adults (50.00%). More than half of the participants had a normal BMI. Rural people with a medium-income range had a more propensity to have depression than others. Unemployment may be a reason to develop MDD because we observed 47.22% of MDD patients were unemployed.

**Table 1. Socio-demographic characteristics of the study population.**

| Characteristics | MDD patients (n = 108) Mean ± SEM | Healthy controls (n = 112) Mean ± SEM | P value |
|---|---|---|---|
| Age in years | 32.15± 0.88 | 33.67± 0.89 | 0.226 |
| 18–30 | 54 (50.00%) | 54 (48.22%) | |
| 31–45 | 45 (41.67%) | 39 (34.82%) | |
| 46–60 | 9 (8.33%) | 19 (16.96%) | |
| Sex | | | 0.513 |
| Male | 52 (48.15%) | 49 (43.75%) | |
| Female | 56 (51.85%) | 63 (56.25%) | |
| Marital Status | | | 0.115 |
| Married | 63 (58.33%) | 71 (63.39%) | |
| Unmarried | 45 (41.67%) | 41 (36.61%) | |
| BMI (kg/m$^2$) | 25.03 ± 0.46 | 24.85 ± 0.38 | 0.761 |
| Below 18.5 (CED) | 3 (2.78%) | 2 (1.79%) | |
| 18.5–25 (normal) | 62 (57.41%) | 60 (53.57%) | |
| Above 25 (obese) | 43 (39.81%) | 50 (44.64%) | |
| Education level | | | < 0.001 |
| No formal education | 13 (12.04%) | 11 (9.82%) | |
| Primary level | 16 (14.82%) | 11 (9.82%) | |
| Secondary level | 31 (28.70%) | 22 (19.64%) | |
| Higher Secondary level | 21 (19.44%) | 25 (22.32%) | |
| Graduate and above | 27 (25%) | 43 (38.40%) | |
| Family income (KBDT)/month | 69.38 ± 3.58 | 77.90± 9.79 | 0.415 |
| Below 40 | 30 (27.78%) | 28 (25%) | |
| 40–100 | 59 (54.63%) | 63 (56.25%) | |
| Above 100 | 19 (17.59%) | 21 (18.75%) | |
| Job status | | | 0.001 |
| Business | 6 (5.56%) | 17 (15.18%) | |
| Service | 18 (16.66%) | 8 (7.14%) | |
| Unemployed | 51 (47.22%) | 36 (32.14%) | |
| Student | 6 (5.56%) | 20 (17.86%) | |
| Others | 27 (25%) | 31 (27.68%) | |
| Economic class | | | 0.892 |
| Low | 30 (27.78%) | 28 (25%) | |
| Medium | 59 (54.63%) | 63 (56.25%) | |
| High | 19 (17.59%) | 21 (18.75%) | |
| Smoking habit | | | 0.848 |
| Yes | 37 (34.25%) | 37 (33.04%) | |
| No | 71 (65.75%) | 75 (66.96%) | |
| Residence area | | | 0.001 |
| Rural | 63 (58.33%) | 42 (37.50%) | |
| Urban | 45 (41.67%) | 70 (62.50%) | |

Abbreviations: BMI, body mass index; CED, chronic energy deficiency; KBDT, kilo Bangladeshi taka; MDD, major depressive disorder; SEM, standard error mean.

## Clinical outcome and laboratory findings

We presented clinical features and laboratory results of the study population in Table 2 and Fig 1. Ham-D scores of MDD patients and HCs were 23.37±0.41 and 5.20±0.31, respectively (p<0.001). Also, DSM-5 scores of MDD patients and HCs were 7.06±0.15 and 1.85±0.16,

**Table 2. Clinical information and laboratory findings of the study population.**

| Parameters | MDD patients (n = 108) Mean ± SEM | Healthy controls (n = 112) Mean ± SEM | *p value* |
|---|---|---|---|
| DSM-5 score | 7.06± 0.15 | 1.85± 0.16 | < 0.001 |
| Ham-D score | 23.37± 0.41 | 5.20 ± 0.31 | < 0.001 |
| Serum resistin level (ng/mL) | 13.82± 1.24 | 6.35± 0.51 | < 0.001 |
| In male (P/C:52/49) | 15.05 ± 2.01 | 6.06 ± 0.79 | < 0.001 |
| In female (P/C:56/63) | 12.68 ±1.50 | 6.57 ± 0.68 | < 0.001 |
| Serum G-CSF level (pg/mL) | 55.45 ± 7.20 | 51.39± 5.89 | 0.660 |
| In male (P/C:52/49) | 64.62 ± 11.33 | 63.92 ± 10.89 | 0.965 |
| In female (P/C:56/63) | 46.93± 8.99 | 41.64± 5.63 | 0.611 |

Abbreviations: DSM-5, diagnostic and statistical manual for mental disorders, 5[th] edition; Ham-D, 17-item Hamilton depression rating scale; G-CSF, granulocyte-colony stimulating factor; MDD, major depressive disorder; P/C, patients and control; SEM, standard error mean.

respectively (p<0.001). We observed higher serum resistin levels in MDD patients (13.82 ±1.24ng/mL) than HCs (6.35±0.51ng/mL). However, we did not observe significant difference in serum G-CSF concentrations between the groups (MDD patients: 55.45±7.20pg/mL; HCs: 51.39±5.89pg/mL; p = 0.660). Moreover, we observed significantly higher levels of serum resistin in both male (15.05±2.01ng/mL) and female (12.68±1.50ng/mL) MDD patients than male (6.06±0.79ng/mL) and female (6.57±0.68ng/mL) in HCs, respectively (Table 2).

In the case of G-CSF, there were no significant changes between the subgroups (male MDD patients and HCs: 64.62±11.33pg/mL and 63.92±10.89pg/mL, respectively, p = 0.965; female MDD patients and HCs: 46.93±8.99pg/mL and 41.64±5.63pg/mL, respectively, p = 0.611).

## Association between clinical outcome and lab findings

Pearson's correlation test established the association between serum resistin levels and Ham-D scores among MDD patients. We observed a significant positive correlation between serum resistin levels and Ham-D scores in MDD patients (r = 0.513, p<0.001). However, we did not find any positive or negative correlations between serum G-CSF levels and Ham-D scores in the patient group. According to sex-specific scatter plot graphs, we noticed female MDD

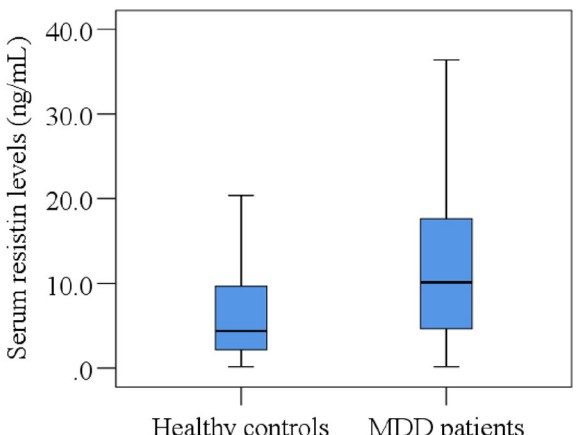
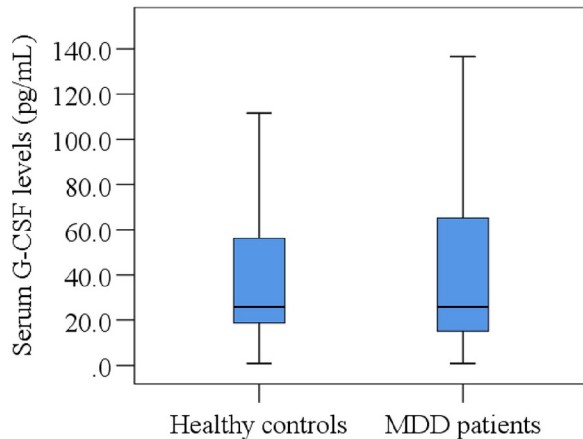

**Fig 1. Distribution of serum inflammatory cytokine levels in MDD patients and healthy controls.** Boxplot graphs showing the median, maximum and minimum value range.

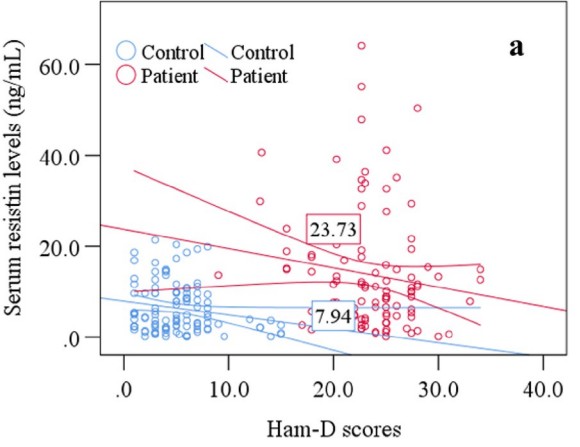 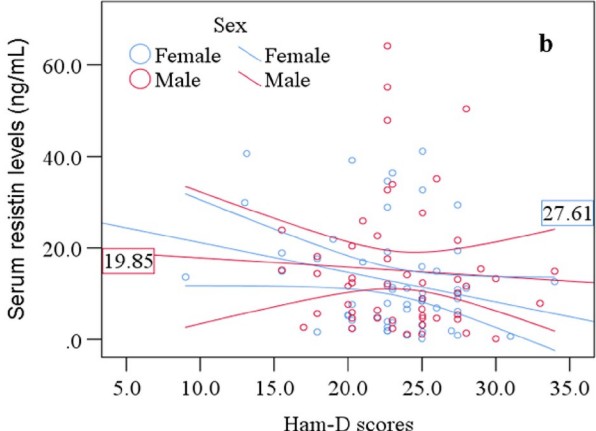

**Fig 2. Scatter plot graphs showing association and mean difference of serum resistin levels with Ham-D scores of study participants.** a: Patient and control specific association, b: Sex specific association among MDD patients.

patients with higher Ham-D scores had elevated serum resistin levels (Fig 2). Again, we did not find such correlations between serum G-CSF levels and Ham-D scores in MDD patients. Also, the ROC curve analysis of serum resistin showed moderate sensitivity and specificity as 65% and 67%, respectively (Fig 3). Moreover, findings from ROC analysis showed the positive prospective value (PPV) and negative prospective value (NPV) as 63% and 62%, respectively. In that case, we detected the cut-off value for serum resistin as 6.39 ng/mL and the area under the curve (AUC) as 0.746. Moreover, we presented the summary of the present study by Fig 4.

## Discussion

The established quantitative diagnostic tests for major depression are still absent. The reason behind the unavailability of laboratory parameters for the diagnosis of depression might be due to insufficient sensitivity and specificity of individual diagnostic markers. The present study investigated serum-based cytokine markers resistin and G-CSF in MDD patients compared to HCs. Here we found elevated serum resistin levels in MDD patients than HCs. We also found a positive correlation between serum resistin levels and Ham-D scores in the patient group. However, serum G-CSF levels did not alter significantly in depression. Also, we found no significant association between serum G-CSF levels and Ham-D scores in MDD patients.

Resistin and G-CSF are inflammatory cytokines, and the actual pathophysiology of these cytokines in the pathogenesis of depression is still elusive. But activation of cytokine receptors in neurons [40], amplified expression of serotonin transporters [41], activation of the kynurenine pathway [42], decreased neuronal growth factors [43], and the stimulation of the HPA axis [44] are possible mechanisms linking cytokine-mediated immune stimulation to the pathogenesis of depression. Moreover, alterations in neurotransmitter synthesis, release, and reuptake can support the above pathophysiological mechanism [45]. Resistin is a pro-inflammatory cytokine produced by adipose tissue that might contribute to insulin resistance in humans [46]. According to some previous study findings, resistin can play its effect on insulin compassion may be mediated via its capability to stimulate the synthesis of anti-inflammatory cytokines (i.e., TNF-α, IL-6) [27,47,48]. A higher level of resistin possesses depressive episodes in MDD as it stimulates the synthesis of anti-inflammatory cytokines such as TNF-α and ketamine [49,50]. However, resistin inhibits dopamine and noradrenaline release in the brain

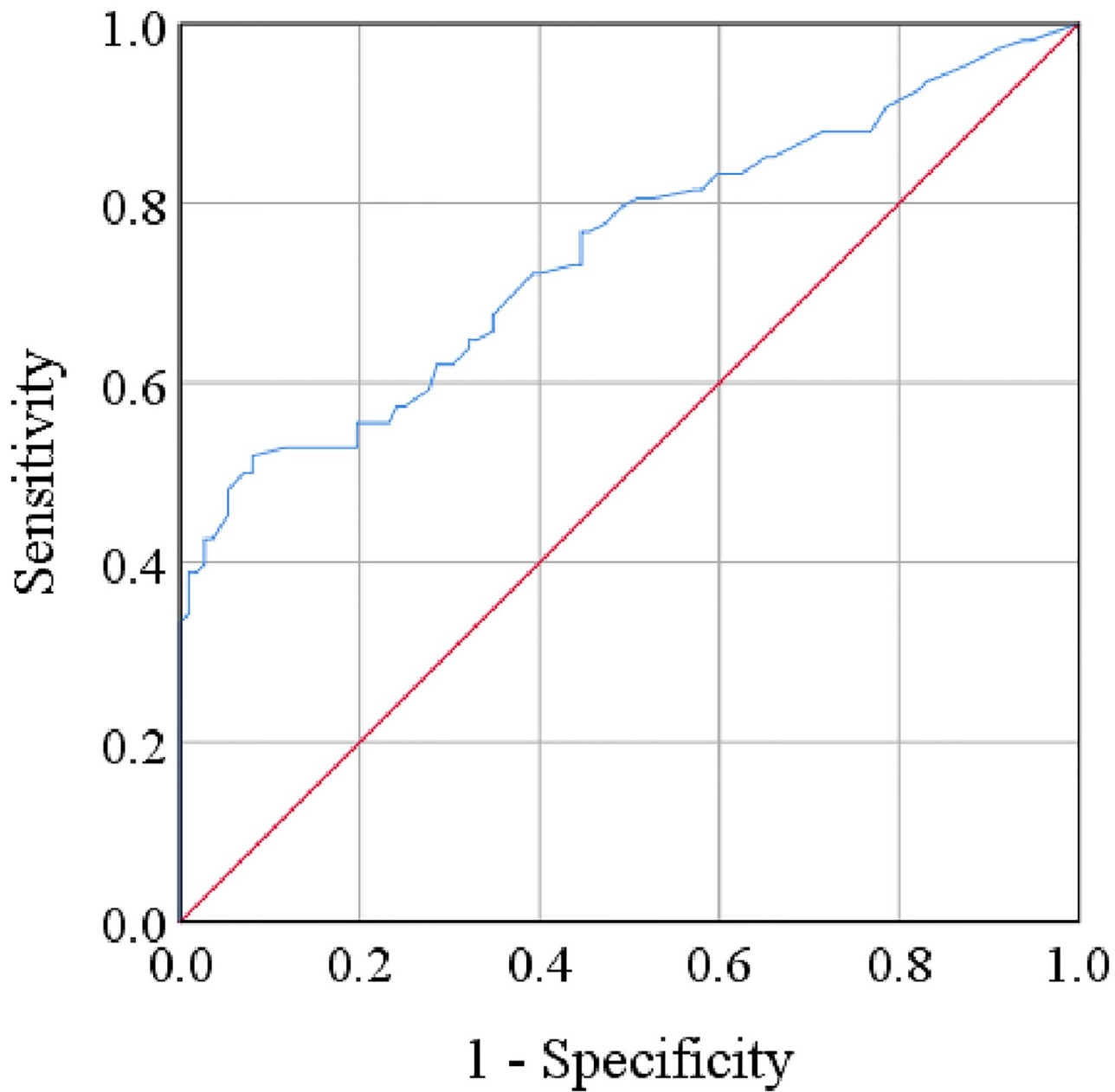

**Fig 3. Receiver operating characteristic (ROC) curve for serum resistin.** The cut-off point was detected as 6.39 ng/mL.

through the hypothalamus. Thus, it can lower the level of intrasynaptic monoamine. Therefore, it can help to develop depressive symptoms in individuals [28]. The research found a positive correlation between circulating levels of resistin and free cortisol concentrations in MDD patients. This association shows that increased serum cortisol is responsible for developing depressive symptoms [9,36]. The finding of the present study is consistent with some previous studies. A study found that serum resistin levels in MDD patients were high before treatment and remained unchanged in individuals whose condition had not improved after antidepressant therapy [36]. Another study found a higher level of resistin in depression. Also, they

**Fig 4. Summary of the study with procedures and findings.**

observed a significant drop in serum resistin levels in patients who had an antidepressant response to ketamine [51]. However, Archer et al. found that women with MDD had greater resistin levels at baseline that did not differ after antidepressant treatment [52]. Jentsch et al. reported higher serum and urine resistin levels in MDD patients. Also, they identified these elevations were more in females than males in a recent gender-specific investigation [53]. Also, another study discovered a positive association between resistin levels and atypical depressive symptoms. However, they did not find any correlations between the typical depressive characteristics and serum resistin levels in MDD patients [21]. Many earlier studies supported their findings [22,27,54–57].

Serum G-CSF levels might involve in the neuroinflammatory process and influence the development of depression. It can also influence leukocyte mobilization from the bone marrow. Therefore, thus increases the production of cytokines [58]. G-CSF helps the propagation and diversity of myeloid progenitors. G-CSF helps the production of granulocyte precursors and inhibition of apoptosis [59,60]. Moreover, G-CSF shows anti-inflammatory and neuroprotective actions in neurodegenerative disorders through dopaminergic neurons in the CNS [29,30]. Therefore, scientists consider G-CSF as a potential medication for neurological diseases [30]. Inconsistent with the present study findings, Lehto et al. did not find any alterations of G-CSF levels in MDD patients compared to HCs [61]. Also, Einvik et al. did not find any significant association between serum level of G-CSF and depression [39]. A recent study

found an inverse relationship between serum G-CSF levels and age in substance use disorder (SUD) compared to HCs [62]. However, Dahl et al. revealed that plasma G-CSF levels became normalized during the recovery phase of depressive episodes in MDD patients [63]. Therefore, we can anticipate that G-CSF might have anti-inflammatory and antidepressant actions based on the above findings [63].

The ROC curve analysis of serum resistin is another strong point of the present study. AUC of ROC analysis was fair (0.746) for serum resisting in depression (p<0.001). Some previous studies measured serum resistin levels in depression, but their results were not robust to evaluate the risk of developing depression due to its low prognostic performance [21,36]. Findings from the ROC analysis showed the sensitivity, specificity, PPV, NPV of serum resistin in MDD patients as 65%, 67%, 63%, and 62% for IL-7, respectively. Similar to the present findings, a longitudinal study reported resistin's association with depressive symptoms. That association was longitudinal, multiply adjusted, and showed a specific mediation of depression's association with cognitive impairment. However, the same study found no association of serum G-CSF with depression [64]. Moreover, major depression is a multifactorial condition accompanied by genetic and environmental features [65]. Therefore, it is vital to define any biological factors precisely that involved might involve in depression. The results of the present investigation showed an increase in serum resistin levels in MDD without a substantial change in G-CSF levels is intriguing.

The present study found elevated serum resistin levels in MDD patients than HCs, but serum G-CSF levels did not alter significantly in depression. Moreover, a positive correlation between serum resistin levels and Ham-D scores in the patient group has been found, but no significant correlation between serum G-CSF levels and Ham-D scores in MDD patients has been established. The findings from the present study might be more complicated than presumed and could vary based on individual MDD patients' characteristics. Therefore, the findings from the study should be considered as preliminary; a new study with a large, homogenous population and by controlling other confounding factors should be conducted in a similar setting to obtain better results. The predictive performance of serum resistin was found fair (0.746) by ROC curve analysis that might be helpful in the management of major depression. Serum resistin might be used to evaluate the risk of developing depression applying the prognostic performance.

## Potential limitations of the study

The present study has a few limitations to consider. The first limitation is that we assessed inflammatory cytokines once during the enrollment of MDD patients and HCs. Secondly, we should consider the effects of dietary supplementation, lifestyle, sleep patterns, and treatment on the investigated parameters in the present investigation. Also, the sample size, case-control nature, unadjusted associations, and modest predictive performance of ROC analysis limit our findings. Therefore, we recommend further studies with more samples and repeated cytokine measurements considering the above factors to produce better results. Despite the above constraints, this study has several strong points. According to our knowledge, it is the first-ever study to examine the association between serum resistin and G-CSF levels among MDD patients in Bangladesh. Also, the present study had a firm match in age and sex between the groups. We followed the same criteria for inclusion and exclusion of both cases and controls. Finally, we analyzed serum resistin and G-CSF levels under the same environment in MDD patients and HCs. Therefore, we hope that the present study findings would help to improve the recently accessible approaches for diagnosing and treating depression.

## Conclusion

Based on the present study findings, patients with severe depression had a considerable increase in serum resistin levels. Therefore, the elevated resistin levels in depression might be the result, not the cause of disease. Nevertheless, we did not find any such alterations in serum G-CSF levels between the groups. We propose serum resistin as a potential candidate marker of depression because of its increasing serum levels, connection with illness severity, and firm diagnostic performance. However, we recommend further studies with larger and more homogeneous samples based on these preliminary study findings.

## Supporting information

**S1 File.**
(XLSX)

## Acknowledgments

All the authors are thankful to participants for their cooperation and participation in this study. We would thank the administrative staff and physicians of the department of psychiatry, Bangabandhu Sheikh Mujib Medical University, Dhaka, Bangladesh. Also, we are thankful to the administration of the University of Asia Pacific for providing reagents and laboratory support.

## Author Contributions

**Conceptualization:** Smaranika Rahman, Amena Alam Shanta, Sardar Mohammad Ashraful Islam, Md. Rabiul Islam.

**Data curation:** Smaranika Rahman, Amena Alam Shanta, Mohammad Shahriar, MMA Shalahuddin Qusar.

**Formal analysis:** Sohel Daria, Zabun Nahar, Mohammad Shahriar, Md. Rabiul Islam.

**Investigation:** Smaranika Rahman, Amena Alam Shanta, Sohel Daria, Md. Rabiul Islam.

**Methodology:** Sardar Mohammad Ashraful Islam, Mohiuddin Ahmed Bhuiyan, Md. Rabiul Islam.

**Project administration:** Mohammad Shahriar, Mohiuddin Ahmed Bhuiyan, Md. Rabiul Islam.

**Supervision:** Md. Rabiul Islam.

**Writing – original draft:** Smaranika Rahman, Sohel Daria.

**Writing – review & editing:** Zabun Nahar, MMA Shalahuddin Qusar, Md. Rabiul Islam.

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
