## [Decision Letter · Decision Letter 0]

6 Dec 2021

PONE-D-21-34686Increased serum resistin but not G-CSF levels are associated in the pathophysiology of major depressive disorder: Findings from a case-control studyPLOS ONE

Dear Dr. Islam,

Thank you for submitting your manuscript to PLOS ONE. After careful consideration, we feel that it has merit but does not fully meet PLOS ONE’s publication criteria as it currently stands. Therefore, we invite you to submit a revised version of the manuscript that addresses the points raised during the review process.

The two reviewers addressed several major and minor concerns about your manuscript. Please revise your manuscript carefully.

We look forward to receiving your revised manuscript.

Kind regards,

Kenji Hashimoto, PhD

Academic Editor

PLOS ONE

Journal Requirements:

Reviewers' comments:

Reviewer's Responses to Questions

**Comments to the Author**

1. Is the manuscript technically sound, and do the data support the conclusions?

Reviewer #1: Yes

Reviewer #2: Yes

2. Has the statistical analysis been performed appropriately and rigorously? 

Reviewer #1: Yes

Reviewer #2: No

3. Have the authors made all data underlying the findings in their manuscript fully available?

Reviewer #1: Yes

Reviewer #2: Yes

4. Is the manuscript presented in an intelligible fashion and written in standard English?

Reviewer #1: Yes

Reviewer #2: Yes

5. Review Comments to the Author

Reviewer #1: Manuscript Number: PONE-D-21-34686

Title: Increased serum resistin but not G-CSF levels are associated in the pathophysiology of major depressive disorder: Findings from a case-control study

Journal: PLOS ONE

MY REPORT

Dear Prof. Dr.

The authors aimed to investigate resistin and G-CSF in MDD patients and controls to explore their role in the pathogenesis and development of depression. I recommend the acceptance of the manuscript, but after a minor revision. The novelty is good. The title and abstract are matched with the rest of the article. The methodology of the study is valid, reliable, and defined appropriately. The data are presented in an appropriate way. However, the discussion, the findings are rather hard to tease out upon a single reading. Can the authors add a diagram/schematic figure that could explain their findings? Also, the authors should add complete sections about the future directions and recommendations. The References are relevant and appropriate for the study. The authors followed the general guidelines of the journal. Finally, the paper will be acceptable for publication if all the above points have been applied.

Thank you very much for your courtesy and time.

With My Kindest and warmest regards

Sincerely yours,

Prof. Dr./Ahmed Ragab Gaber (Ahmed R. G.)

Zoology department, Faculty of science

Beni-suef University, Egypt

E-mails; ahmedragab08@gmail.com

Mobil phone: +02-01091471828.

Reviewer #2: RE: PONE-D-21-34686

The authors tested the association between clinical diagnoses of major depressive disorder (MDD) and depressive symptom rating scales, and two blood-based biomarkers measured in serum: resistin and granulocyte colony-stimulating factor (G-CSF). Only resistin levels were elevated in MDD patients compared with controls. Scores on the Hamilton Depression Rating Scale (Ham-D) significantly and moderately strongly correlated with serum resistin levels. The association was unadjusted. Serum resistin had a modest predictive association with MDD by Receiver Operating Characteristic (ROC) analysis (AUC = 0.75).

There is little new in this analysis which is cross-sectional, in a small number of cases, and unadjusted. Only two cytokines are reported, making resistin’s unique association difficult to interpret. Both resistin’s association with depressive symptoms and G-CSF’s lack of association have been previously reported in this journal, [1] and yet that reference is not cited. That association was longitudinal, multiply adjusted, and showed a specific mediation of depression’s association with cognitive impairment, and that too was not discussed. Figure 2 is almost uninterpretable.

The axes should be reversed and regressions fit to the entire sample. The gender split adds little. There was no test of significance but it looks insignificant.

This analysis is better suited for a letter or brief report of a confirmatory finding.

1. Royall DR, Al-Rubaye S, Bishnoi R, Palmer RF. Serum protein mediators of depression’s association with dementia. PLoS One. 2017;12:e0175790.

6. PLOS authors have the option to publish the peer review history of their article (what does this mean?). If published, this will include your full peer review and any attached files.

Reviewer #1: No

Reviewer #2: No

---

## [Author Response · Author response to Decision Letter 0]

15 Dec 2021

Dear Editors and Reviewers,

Thank you for your letter and the reviewers' comments on our manuscript entitled "Increased serum resistin but not G-CSF levels are associated in the pathophysiology of major depressive disorder: Findings from a case-control study" (Manuscript ID PONE-D-21-34686). All the comments were valuable and helpful to the revision and improvement of the manuscript. We have carefully studied the comments and made corrections, which we hope will merit your approval. We marked the revised portions using track changes. Our point-by-point answers to the reviewers’ comments appear at the end of this letter.

We earnestly appreciate the Editors'/Reviewers' work. We hope that after this revision, the paper will be deemed fit for publication. We would be glad to respond to any further questions and comments that you may have. 

Once again, thank you very much for your comments and suggestions.

Best regards,

Md. Rabiul Islam, PhD

Assistant Professor, Department of Pharmacy, University of Asia Pacific, 74/A Green Road, Farmgate, Dhaka-1215, Bangladesh. Email: robi.ayaan@gmail.com; Cell: +8801916031831

Point by point authors’ responses to the reviewers

Manuscript ID PONE-D-21-34686

Title: Increased serum resistin but not G-CSF levels are associated in the pathophysiology of major depressive disorder: Findings from a case-control study

Reviewer #1

Comment to author:

Dear Prof. Dr.

The authors aimed to investigate resistin and G-CSF in MDD patients and controls to explore their role in the pathogenesis and development of depression. I recommend the acceptance of the manuscript, but after a minor revision. The novelty is good. The title and abstract are matched with the rest of the article. The methodology of the study is valid, reliable, and defined appropriately. The data are presented in an appropriate way. However, the discussion, the findings are rather hard to tease out upon a single reading. Can the authors add a diagram/schematic figure that could explain their findings? Also, the authors should add complete sections about the future directions and recommendations. The References are relevant and appropriate for the study. The authors followed the general guidelines of the journal. Finally, the paper will be acceptable for publication if all the above points have been applied.

Thank you very much for your courtesy and time.

With My Kindest and warmest regards

Sincerely yours,

Author responses

Thank you for your review and valuable observation. We appreciate your encouraging comments on our manuscript. 

Following your suggestion, we have added a diagram about the summary of this study that we believe would help the readers to understand the present study and its findings easily. This diagram can be viewed as Figure 4. 

Also, we added a separate section in the revised version regarding the future directions and recommendations based on the present study findings (page 15, line 298-304, page 16, line 305-308, in revised version).

Reviewer #2

Comment to author:

The authors tested the association between clinical diagnoses of major depressive disorder (MDD) and depressive symptom rating scales, and two blood-based biomarkers measured in serum: resistin and granulocyte colony-stimulating factor (G-CSF). Only resistin levels were elevated in MDD patients compared with controls. Scores on the Hamilton Depression Rating Scale (Ham-D) significantly and moderately strongly correlated with serum resistin levels. The association was unadjusted. Serum resistin had a modest predictive association with MDD by Receiver Operating Characteristic (ROC) analysis (AUC = 0.75).

There is little new in this analysis which is cross-sectional, in a small number of cases, and unadjusted. Only two cytokines are reported, making resistin’s unique association difficult to interpret. Both resistin’s association with depressive symptoms and G-CSF’s lack of association have been previously reported in this journal, [1] and yet that reference is not cited. That association was longitudinal, multiply adjusted, and showed a specific mediation of depression’s association with cognitive impairment, and that too was not discussed. Figure 2 is almost uninterpretable.

The axes should be reversed and regressions fit to the entire sample. The gender split adds little. There was no test of significance but it looks insignificant.

This analysis is better suited for a letter or brief report of a confirmatory finding.

1. Royall DR, Al-Rubaye S, Bishnoi R, Palmer RF. Serum protein mediators of depression’s association with dementia. PLoS One. 2017;12:e0175790. 

Author responses

Thank you for your observation. The small sample size, case-control nature, unadjusted associations, and modest predictive performance of ROC of the present study have been mentioned as potential limitations in the revised manuscript (page 16, line 313-314). Also, we suggested future directions and recommendations based on the present findings (page 15, line 298-304, page 16, line 305-308, in revised version).

Following your observation and suggestion, we have now discussed the suggested literature in the revised manuscript. We believe this discussion would add value to the present study (page 15, 289-293). Also, we believe resistin levels in y-axis and severity scale in x-axis seem okay to show the associations between them (Figure 2).

---

## [Decision Letter · Decision Letter 1]

10 Feb 2022

Increased serum resistin but not G-CSF levels are associated in the pathophysiology of major depressive disorder: Findings from a case-control study

PONE-D-21-34686R1

Dear Dr. Islam,

We’re pleased to inform you that your manuscript has been judged scientifically suitable for publication and will be formally accepted for publication once it meets all outstanding technical requirements.

Kind regards,

Kenji Hashimoto, PhD

Section Editor

PLOS ONE

Additional Editor Comments (optional):

Reviewers' comments:

Reviewer's Responses to Questions

**Comments to the Author**

1. If the authors have adequately addressed your comments raised in a previous round of review and you feel that this manuscript is now acceptable for publication, you may indicate that here to bypass the “Comments to the Author” section, enter your conflict of interest statement in the “Confidential to Editor” section, and submit your "Accept" recommendation.

Reviewer #1: (No Response)

2. Is the manuscript technically sound, and do the data support the conclusions?

Reviewer #1: (No Response)

3. Has the statistical analysis been performed appropriately and rigorously? 

Reviewer #1: (No Response)

4. Have the authors made all data underlying the findings in their manuscript fully available?

Reviewer #1: (No Response)

5. Is the manuscript presented in an intelligible fashion and written in standard English?

Reviewer #1: (No Response)

6. Review Comments to the Author

Reviewer #1: (No Response)

7. PLOS authors have the option to publish the peer review history of their article (what does this mean?). If published, this will include your full peer review and any attached files.

Reviewer #1: **Yes: **Ahmed R.G., Faculty of Science, Beni-Seuf University

---

## [Editor Report · Acceptance letter]

15 Feb 2022

PONE-D-21-34686R1 

Increased serum resistin but not G-CSF levels are associated in the pathophysiology of major depressive disorder: Findings from a case-control study 

Dear Dr. Islam:

I'm pleased to inform you that your manuscript has been deemed suitable for publication in PLOS ONE. Congratulations! Your manuscript is now with our production department. 

Kind regards, 

on behalf of

Prof. Kenji Hashimoto 

Section Editor

PLOS ONE